# Molecular requirements for *C. elegans* transgenerational epigenetic inheritance of pathogen avoidance

**Rachel Kaletsky**[1,2], **Rebecca S Moore**[1,2,3,4], **Titas Sengupta**[1,2], **Renee Seto**[1,2], **Borja Ceballos-Llera**[1,2], **Coleen T Murphy**[1,2]*

[1]Department of Molecular Biology, Princeton University, Princeton, United States; [2]LSI Genomics, Princeton University, Princeton, United States; [3]Chronobiology and Sleep Institute, Department of Neuroscience, Philadelphia, United States; [4]Howard Hughes Medical Institute, University of Pennsylvania Perelman School of Medicine, Philadelphia, United States

*For correspondence:
ctmurphy@princeton.edu

**Competing interest:** The authors declare that no competing interests exist.

## eLife Assessment

This **fundamental** study concerns a model for transgenerational epigenetic inheritance, the learned avoidance by *C. elegans* of the PA14 pathogenic strain of *Pseudomonas aeruginosa*. The authors test the impact of procedural alterations made in another study, by Gainey et al., which claimed that transgenerational inheritance in this paradigm lacks robustness, despite this observation having been reported in multiple papers from the Murphy lab. The authors of the present study show that by following a non-standard avoidance protocol, Gainey et al. likely biased their measurements in a way that made it hard to observe learned avoidance. The authors also highlight the importance of bacterial growth conditions, showing that expression of the trigger molecule, the bacterial P11 RNA, which is necessary and sufficient to drive the transgenerational inheritance of the avoidance phenotype, is influenced by temperature. As expression of P11 was not verified by Gainey et al., this provides another explanation for the inability to observe transgenerational epigenetic inheritance. Together, the authors provide **compelling** and powerful arguments that the original phenomenon is robust and that it can be reproduced in the Murphy lab by following their original protocol precisely, including the use of azide to immobilize the worms at the food source. Overall, this study not only provides guidance for investigators in this experimental paradigm, but it also provides additional understanding of the differences between naïve preference, learned preference, and transgenerational epigenetic inheritance. The present study is therefore of broad interest to anyone studying genetics, epigenetics, or learned behavior.

**Abstract** Bacteria are *Caenorhabditis elegans'* food, and worms are naturally attracted to many bacteria, including pathogenic *Pseudomonas*, preferring PA14 over laboratory *Escherichia coli* (OP50). Despite this natural attraction to PA14, prior PA14 exposure causes the worms to instead avoid PA14. This behavioral switch can happen quickly – even within the duration of the choice assay. We show that accurate assessment of the animals' true first choice requires the use of a paralytic (azide) to trap the worms at their initial choice, preventing the switch from attraction to avoidance of PA14 within the assay period. We previously discovered that exposure of *C. elegans* to 25°C plate-grown PA14 at 20°C for 24 hr not only leads to PA14 avoidance, but also to four generations of naïve progeny avoiding PA14, while other PA14 paradigms only cause P0 and/or F1 avoidance. We also showed that the transgenerational (P0-F4) epigenetic avoidance is mediated by P11, a small RNA produced by PA14. P11 is both necessary and sufficient for TEI of learned avoidance. P11 is

highly expressed in our standard growth conditions (25°C on surfaces), but not in other conditions, suggesting that the reported failure to observe F2-F4 avoidance is likely due to the absence of P11 expression in PA14 in the experimenters' growth conditions. Additionally, we tested ~35 genes for involvement in TEI of learned pathogen avoidance. The conservation of multiple components of this sRNA TEI mechanism across *C. elegans* strains and in multiple *Pseudomonas* species suggests that this TEI behavior is likely to be physiologically important in wild conditions.

## Introduction

*Caenorhabditis elegans* in the wild is found primarily in rotting fruit, where a range of bacterial species serve as abundant food sources (*Félix and Duveau, 2012*). About a third of these species are of the *Pseudomonas* family (*Samuel et al., 2016*), and many of those *Pseudomonas* species are pathogenic to *C. elegans*. Therefore, the development of multiple strategies to detect and avoid these pathogens that act at different timescales is critical for *C. elegans'* fitness and survival.

*C. elegans* and a clinical isolate of *Pseudomonas aeruginosa*, PA14, have been studied as a model of host–pathogen interactions for several decades (*Tan et al., 1999*). PA14 kills *C. elegans* through several distinct mechanisms (e.g. slow and fast killing [*Mahajan-Miklos et al., 1999*]) and typically *C. elegans* adults may only survive for a few days on PA14 (*Tan et al., 1999*), cutting its reproductive period short. *C. elegans'* survival depends on the expression of antimicrobial genes that are induced downstream of innate immune signals in response to PA14 exposure (*Troemel et al., 2006*) and changes to developmental rates and decisions that increase the odds of the next generation's survival.

In addition to the activation of pathogen survival genes and effects on reproduction and development, *C. elegans* also changes its behavior in the presence of PA14. Despite its potential to kill *C. elegans,* worms are initially attracted to PA14 (*Zhang et al., 2005*), because *C. elegans* cannot distinguish pathogenic from nonpathogenic *Pseudomonas* species that may serve as nutritious food sources (*Sengupta et al., 2024*). Specific mechanisms to detect, leave, and avoid PA14 are activated via secreted bacterial metabolites (*Meisel et al., 2014*; *Prakash et al., 2021*) or through internal signals that indicate illness (*Melo and Ruvkun, 2012*; *Singh and Aballay, 2019*; *Zhang et al., 2005*). Another mechanism of learned avoidance can be activated upon exposure to pathogenic *Pseudomonas* and is mediated by specific bacterial small RNAs (*Moore et al., 2019*; *Kaletsky et al., 2020*; *Sengupta et al., 2024*; *Seto et al., 2025*). These distinct avoidance pathways are triggered by specific molecular cues, are independently regulated, and act on different timescales. Specifically, PA14 metabolites induce a distinct, ASJ-mediated avoidance pathway in P0s (*Meisel et al., 2014*); 4 hr of PA14 exposure induces avoidance directly in the worms and increased attraction in the F1s, while 8 hr of treatment can induce F1 avoidance but does not persist beyond F1 (*Pereira et al., 2020*; *Zhang et al., 2005*), and specific bacterial small RNAs from PA14 as well as from wild *Pseudomonas* species induce avoidance after 24 hr of treatment that is maintained through four generations of progeny (*Kaletsky et al., 2020*; *Moore et al., 2019*; *Sengupta et al., 2024*; *Seto et al., 2025*). Through extensive mutant analyses, we have found that this transgenerational avoidance mechanism requires a wide range of machinery, including RNA interference components, germline components, the Cer1 Ty3 Gypsy retrotransposon, and neuronal function (*Moore et al., 2019*; *Kaletsky et al., 2020*; *Moore et al., 2021*; *Sengupta et al., 2024*; *Seto et al., 2025*).

Here, we discuss the molecular conditions that are necessary to induce the bacterial small RNA-induced pathway of transgenerationally inherited learned avoidance behavior. Our results suggest that one major difference in the execution of the assay itself, the presence or absence of the paralytic sodium azide, may account for some reported discrepancies in naïve and learned chemotaxis results (*Gainey et al., 2024*). Perhaps more importantly, PA14 must be grown under conditions that induce the expression of the small RNA, P11, which is necessary and sufficient for transgenerational inheritance of learned pathogen avoidance; P11 regulates nitrogen fixation in PA14, and these conditions promote biofilm formation and thus may be important signals for *C. elegans* to detect the pathogen. These two factors, P11 expression and azide presence in the assay, are critical in obtaining consistent naïve and learned avoidance results. Together with the identification of genes and pathways that are involved, we are beginning to understand the molecular requirements for transgenerational epigenetic inheritance of learned avoidance.

## Results

### Choice assays require a paralytic to accurately capture first choice

Because *C. elegans* has several different mechanisms of pathogen avoidance that can be detected via different signals (olfactory cues, gustatory cues, illness-induced signals, bacterial metabolites, bacterial RNAs, etc.), pathogen avoidance behavior has been tested using several different assays (as reviewed in *Liu and Zhang, 2020*). Among these are variations on the bacterial choice assay, which is modeled on standard chemotaxis choice assays (two small bacterial spots with azide to trap worms at their first choice) (*Bargmann et al., 1993*; *Hart, 2006*; *Zhang et al., 2005*; *Jin et al., 2016*; *Lee and Mylonakis, 2017*; *Ooi and Prahlad, 2017*; *Liu et al., 2018*; *Moore et al., 2019*), variations on single-worm head-turning assays (*Ha et al., 2010*), and the leaving assay (PA14 spot in middle of plate, no other bacteria, no azide, and monitored over long periods for worms leaving the spot, e.g. 8–24 hr) (*Meisel et al., 2014*; *Miller et al., 2015*; *Ma et al., 2017*; *Horspool and Chang, 2017*; *Hao et al., 2018*; *Singh and Aballay, 2019*; *Wolfe et al., 2019*; *Filipowicz et al., 2022*). Each assay measures responses on a specific timescale; e.g., the two-bacteria choice assay measures initial preference of a population of animals over an hour, the head-turning assay measures immediate preference of individual animals, and the leaving assay (which is often performed overnight) measures final preference over many hours of PA14 exposure. Therefore, the conditions used in one assay might not be appropriate for all assays.

The standard bacterial choice assay is performed on plates, with two small spots of bacteria (here, PA14 and OP50) separated by several centimeters that the worms must traverse in response to olfactory cues (*Figure 1A*); monitoring the accumulation of worms as they become paralyzed with the anesthetic sodium azide at each bacterial spot allows the experimenter to accurately assess the worm's first choice. Studies by many groups, including *Zhang et al., 2005*; *Meisel et al., 2014*; *Ooi and Prahlad, 2017*; *Pereira et al., 2020*; *Filipowicz et al., 2022*, and others over the past two decades have previously demonstrated that *C. elegans* are initially attracted to PA14 without prior exposure to the pathogen ('naïve') but switch their behavior to avoidance of PA14 following exposure to PA14. Our lab has also consistently replicated those naïve attraction and learned avoidance results: in fact, we see that the mild naïve attraction to PA14 is fairly consistent between individual plates and individual replicates over many experiments across several studies (*Figure 1B–D*, *Supplementary file 1*; *Moore et al., 2019*; *Kaletsky et al., 2020*; *Moore et al., 2021*; *Sengupta et al., 2024*; *Seto et al., 2025*). This initial naïve attraction to PA14 serves as a necessary baseline for all further experiments. That is, if no naïve attraction to PA14 is observed and instead the worms already avoid PA14 prior to PA14 exposure, it might be difficult to observe any additional learned avoidance of PA14 after training. Failure to observe naïve attraction to PA14 might also suggest that the experimenters' assay conditions are not the same as those used by previous groups. A recent study failed to consistently observe this initial naïve avoidance of PA14 (*Gainey et al., 2024*); therefore, we investigated possible origins of this deviation in these results from other groups' previously published results.

To determine possible origins of deviations in naïve choice assay results, we carried out the two-bacteria choice assay under standard conditions (1 hr of a binary choice of PA14 vs OP50, which includes the paralytic sodium azide added to the bacterial spots) as well as the conditions reported by Gainey et al., which primarily differs by omitting azide and letting the worms choose in the absence of azide for 1 hr at room temperature, and then moving the plates to cold temperature (4°C) prior to counting, as described by *Gainey et al., 2024*. We found that when azide is present at the bacterial spots to trap naïve worms at their first choice, the worms demonstrate attraction to PA14, as has been previously shown, but in the absence of azide, worms show significant avoidance of PA14 even without prior exposure to the pathogen (*Figure 1E and F*); the difference between the +azide and -azide chemotaxis indices is both significant and large in magnitude (a difference of 0.57 in CI and p<0.0001; *Figure 1F*). We note that these no-azide results are almost identical to the results that Gainey et al. reported for their naive P0s, while the +azide results are similar to our previously published naïve data (*Figure 1B–D*), as well as published data from other labs.

To better understand the source of the discrepancy with the Gainey et al. results, we first investigated the worms' behavior at 4°C after the hour of the assay at room temperature; while we do see that the worms continue to move (4°C does not immediately paralyze worms), in fact, the final choice index of the population was already reached at the end of the room temperature period *before* they were placed in the cold (*Figure 1G*, t=0); therefore, the discrepancy in attraction to/avoidance of

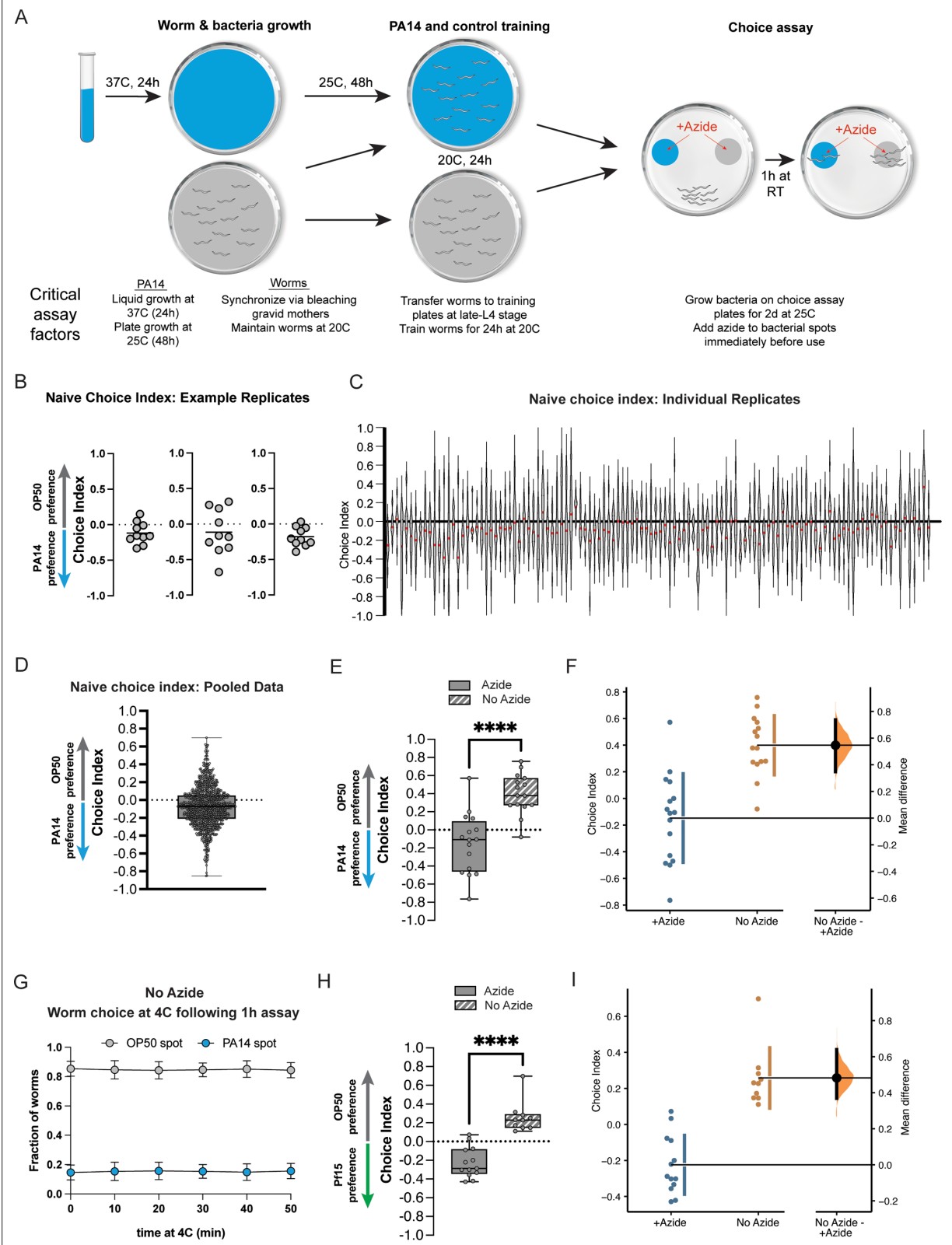

**Figure 1.** The omission of the paralytic sodium azide from choice assays has significant effects on naïve choice assays. (**A**) Schematic of PA14 and worm growth conditions, worm training, and choice assay parameters. Critical factors affecting assay performance are highlighted. (**B**) Scatter dot plots of representative replicate choice assays (OP50 vs PA14 bacteria) from naïve day 1 adult worms (from *Moore et al., 2019*; *Kaletsky et al., 2020*; *Moore et al., 2021*). Each dot represents an individual choice assay plate (n=10) containing ~50–100 worms per plate (average ~80). Choice index = (number

*Figure 1 continued*

of worms on OP50 − number of worms on PA14)/(total number of worms). (**C**) Individual replicates of naïve choice assays from *Moore et al., 2019*; *Kaletsky et al., 2020*; *Moore et al., 2021*, are shown as violin plots. Each replicate is a separate experiment made up of 5–10 plates. The median is shown in red. (**D**) Individual replicates from (**C**) were pooled. Box plots: center line, median; box range, 25–75th percentiles; whiskers denote minimum–maximum values. (**E**) Naïve worms were placed on choice assay plates with spots of PA14 and OP50 with or without sodium azide on the bacterial spots, then allowed to choose for 1 hr, followed by 1 hr at 4°C. No azide: CI = 0.4 ± 0.06, p<0.0001. (**F**) Estimation plot of difference in CI between azide and no azide plates; mean difference = 0.482 [95.0%CI 0.369, 0.64]. (**G**) After 1 hr of the choice assay at room temperature, choice assay plates were placed at 4°C. Worms were counted at each spot at the indicated time points. Mean ± SD from three choice assay plates. Time point 0=immediately before 4°C incubation. (**H**) Worms were placed on choice assay plates with spots of *Pseudomonas fluorescens 15* (PF15) and OP50 with or without sodium azide on the bacterial spots, then allowed to choose for 1 hr, followed by 1 hr at 4°C. No azide CI = 0.26 ± 0.05, p<0.0001. (**i**) Estimation plot of difference in CI between azide and no azide plates; mean difference = 0.547 [95.0%CI 0.348, 0.737].

PA14 seems to have occurred during the initial part of the assay, when the worms were choosing for 1 hr at room temperature without azide. This may be particularly relevant for the Gainey et al. study, as they reportedly moved the worms at different times ('*These no-azide assay plates were moved to 4°C after 30–60* min'), which likely induced further variance. We also observed worms still traveling away from the PA14 spot at 1 hr, further underscoring the point that performing the assay without azide fails to trap the worms at their first choice, even if the worms are transferred to 4°C, because the worms have already made the switch from attraction to avoidance by that point.

The simplest interpretation of the discrepancy between the ±azide assay conditions is that the worms are initially attracted to and choose PA14; in the presence of azide, the worms remain at this first-choice spot, but over the course of an hour-long choice assay in the absence of azide, the worms learn to avoid PA14 and migrate to the OP50 spot. This phenomenon was already carefully demonstrated by *Ooi and Prahlad, 2017*, with a time-course assay: in the absence of azide to trap the worms, the worms are initially attracted to PA14, but by 45 min the worms show no preference for PA14, and by 1 hr the worms instead avoid PA14 – i.e., within the time of the hour-long assay, in the absence of azide to trap the worms at their first choice, they learn to avoid PA14. Under the no-azide assay conditions, it would be difficult to observe any additional learned avoidance that is the result of training, since the learned preference for OP50 (PA14) is already very high when the worms have been allowed to change from their first choice to their second choice. Therefore, the simplest interpretation is that one cannot accurately measure naïve preference in the absence of azide, and one cannot subsequently properly assess subsequent learned avoidance after training without azide. It is notable that the overwhelming majority of Gainey et al.'s experiments were done without azide; because a third party (*Ooi and Prahlad, 2017*) already showed that there is a substantial difference when no azide is used, as we have shown here, Gainey et al.'s no-azide data cannot be compared with assays done with azide, including all of our work.

We repeated this ±azide preference assay with another *Pseudomonas* pathogen, *P. fluorescens 15* (PF15), that we have also found to induce transgenerational avoidance for five generations via small RNAs (*Seto et al., 2025*) and observed similar results (*Figure 1H and I*). That is, the preference for PF15 is similar to worms' preference for PA14 over OP50, but in the absence of azide, the worms learn to avoid PF15, as they did with PA14. Therefore, we conclude that the inclusion of azide in the choice assay is necessary to properly assess the worms' first choice; in the absence of azide, the worms' naïve choice is not being measured, so it cannot be used to then compare with post-training behavior to assess learned or inherited avoidance.

## Transgenerational inheritance of learned avoidance is consistent and distinct from other avoidance paradigms

*C. elegans* uses many different environmental cues to inform its behavioral decisions; in fact, avoidance of PA14 can be induced by bacterial metabolites, by olfactory cues, by gustatory cues, and by innate immunity pathways. These cues induce avoidance in the P0 generation, and they and other mechanisms can induce avoidance and changes in survival or development in the F1 generation via distinct mechanisms (*Pereira et al., 2020*; *Hong et al., 2021*; *Burton et al., 2020*; *Pender et al., 2025*), but do not extend beyond the F1 generation. Our studies used a single condition for training, PA14 grown on plates at 25°C and training at 20°C for 24 hr (see *Figure 1A*). Under these conditions, we consistently observe learned avoidance in mothers (P0; *Figure 2A, B, and C*) that is then

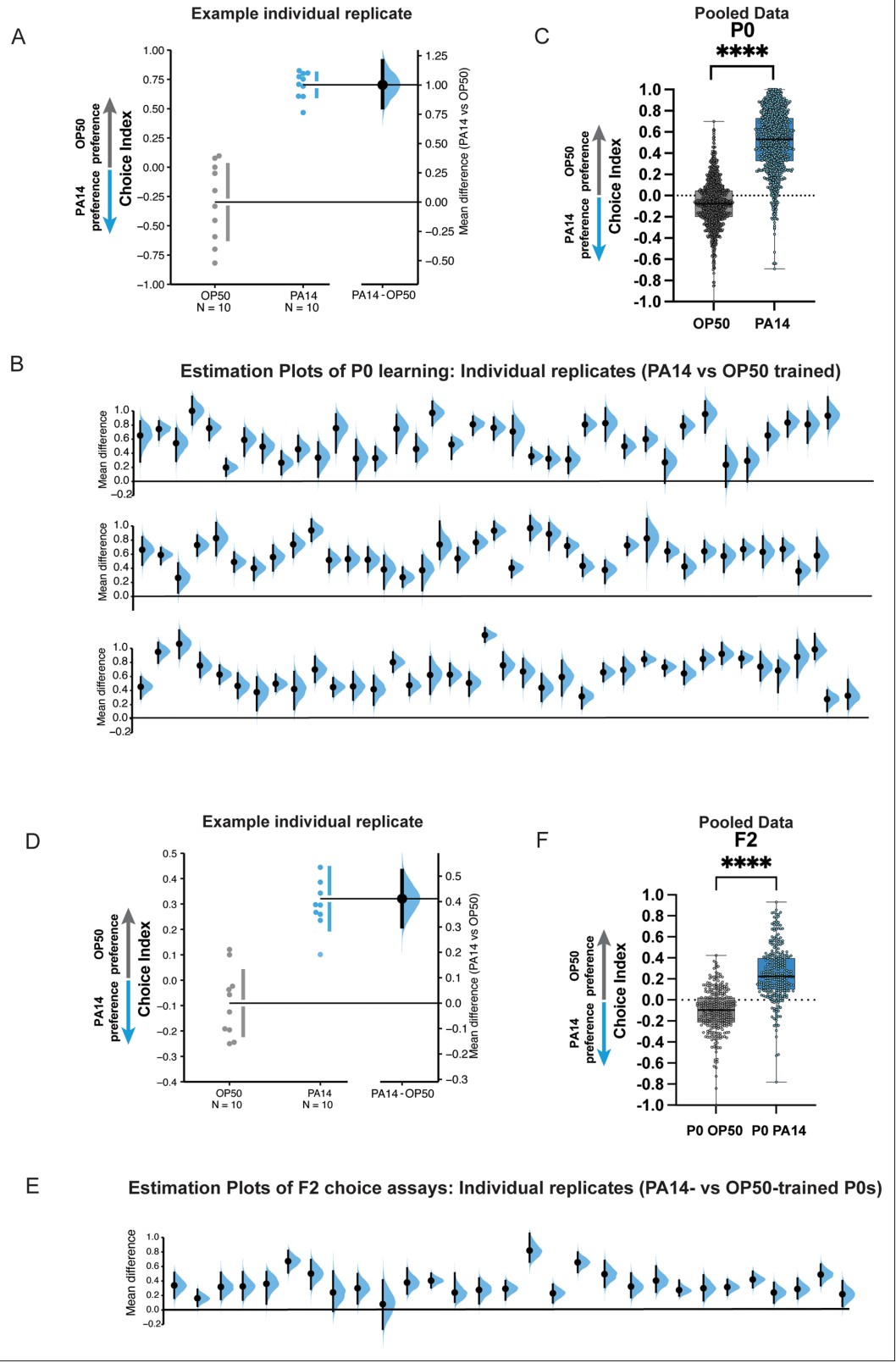

**Figure 2.** P0 and F2 worms trained on PA14 bacteria reproducibly learn to avoid PA14 in choice assays. (**A**) P0 replicates: A representative experiment (replicate) showing the choice index and effect size of 24 hr PA14-trained worms compared to the OP50-trained control. (**B**) Mean differences of individual replicates from *Moore et al., 2019*; *Kaletsky et al., 2020*; *Moore et al., 2021*, are shown for PA14- vs OP50-trained mothers. (**C**) Individual

*Figure 2 continued on next page*

*Figure 2 continued*

replicates from (**B**) were pooled. (**D–E**) F2 animals from P0-control or PA14-trained worms. (**D**) F2 replicates: A representative experiment (replicate) showing the choice index and effect size of F2 worms. (**A, D**) The mean difference is shown as a Gardner–Altman estimation plot. Both groups are plotted on the left axes; the mean difference is plotted on floating axes on the right as a bootstrap sampling distribution. The mean difference is depicted as a dot; the 95% confidence interval is indicated by the ends of the vertical error bar. (**E**) Mean differences of individual replicates from *Moore et al., 2019*; *Kaletsky et al., 2020*; *Moore et al., 2021*, are shown for F2s from PA14- vs OP50-trained grandmothers. (**B, E**) Box plots: center line, median; box range, 25–75th percentiles; whiskers denote minimum–maximum values. Unpaired, two-tailed Student's t-test. ****p<0.0001. (**F**) Individual replicates from (**E**) were pooled. (**C, F**) Each replicate is a separate experiment made up of 5–10 plates (average ~80 worms per plate). The mean difference (effect size) is shown as a Cumming estimation plot. Each mean difference is plotted as a bootstrap sampling distribution. Mean differences are depicted as dots; 95% confidence intervals are indicated by the ends of the vertical error bars. Estimation graphics generated as described in *Ho et al., 2019*.

transgenerationally inherited (F2; *Figure 2D–F*). Therefore, our PA14 growth and/or training conditions must be distinct from those growth and training paradigms that only produce P0 or F1 effects.

## P11 expression is condition-specific and critical for TEI of learned avoidance

In searching for the relevant TEI-inducing factor, we tested and eliminated DNA, proteins, and large RNAs as the TEI-inducing component and found that small RNAs isolated from PA14 grown at 25°C on plates could induce avoidance and TEI (*Kaletsky et al., 2020*). That is, small RNAs isolated from 25°C plate-grown PA14 pipetted onto *Escherichia coli* prior to 24 hr *C. elegans* training were able to induce avoidance in those animals and the transgenerational inheritance of that avoidance, just as we had observed from 24 hr PA14 lawn-training experiments. We had also noticed that PA14 grown at lower temperatures or in liquid could not induce transgenerational inheritance of learned avoidance. Capitalizing on these observations, we used differential small RNA sequencing to identify the set of small RNAs that are expressed in PA14 grown at 25°C on plates that are not expressed in liquid or at low temperatures; we then cloned each of those six small RNAs into *E. coli* and tested these individual sRNAs for their ability to induce avoidance and TEI (*Kaletsky et al., 2020*). This set of experiments revealed that the small RNA P11 – and only P11 – could induce avoidance in mothers and inheritance of avoidance for four generations (*Kaletsky et al., 2020*). *C. elegans* trained on *E. coli* expressing P11 sRNA induced the P0-F4 avoidance of PA14 without making the worms sick and without inducing innate immune pathways, but while also inducing *daf-7::gfp* expression in the ASI (*Kaletsky et al., 2020*). We observed that *E. coli*-P11 (indicated by the yellow points) consistently induces P0 avoidance (*Figure 3A and B*), and this learned avoidance is inherited transgenerationally (*Figure 3C and D*). Therefore, P11 sRNA is both necessary and sufficient to induce the transgenerational inheritance of learned PA14 avoidance.

We noted that P0 avoidance induced by *E. coli*-P11 training (*Figure 3A and B*) is slightly lower than that induced by training on PA14 bacterial lawns (*Figure 2B and C*), suggesting that the P0 innate immunity-induced and other avoidance mechanisms are distinct from and additive with the sRNA pathway in mothers. In fact, the average level of avoidance induced by *E. coli*-P11 training is consistent in P0 and F2 (*Figure 3C and D*), and with the F2 avoidance levels induced by PA14 training (*Figure 2E and F*), consistent with our previous reports (*Kaletsky et al., 2020*; *Moore et al., 2019*). P11-induced avoidance remains consistent from P0 through F4 (*Kaletsky et al., 2020*), as we later also found for the small RNAs (Pv1, Pfs1) required for *Pseudomonas vranovensis* (*Sengupta et al., 2024*) and PF15 (*Seto et al., 2025*) avoidance. Similarly, while *E. coli*-P11 training induced *daf-7::gfp* expression in the ASI from P0 through F4 (*Kaletsky et al., 2020*), we did not observe any *daf-7::gfp* expression in the ASJ, consistent with the ASJ response being specifically induced by metabolites produced by PA14 (*Meisel et al., 2014*). Therefore, it appears that there are PA14 growth conditions that can induce some mechanisms of P0 and F1 avoidance, but will not induce transgenerational avoidance, because no P11 is expressed in PA14 under those conditions.

To more carefully address this hypothesis, we examined the levels of P11 sRNA from PA14 grown at 25°C on plates, 15°C on plates, and 37°C in liquid (*Figure 3E*, adapted from *Kaletsky et al., 2020*). When comparing 25°C plates with 15°C plates, we observed a 6.8-fold decrease in P11 levels at

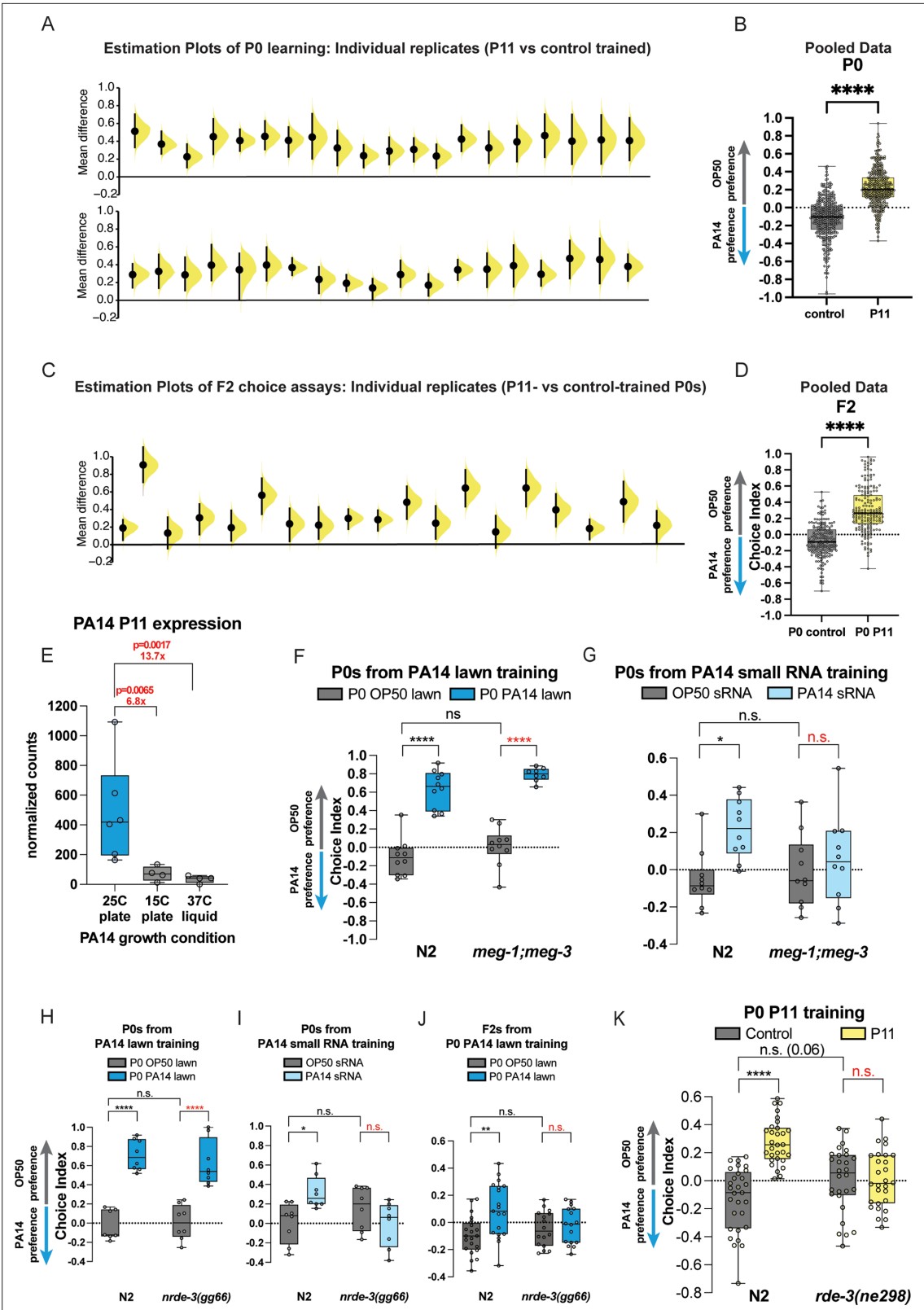

**Figure 3.** P11 sRNA and components of germline and RNAi function are required for transgenerational inheritance. (**A**) P0 worms trained on *E. coli* expressing the P11 small RNA reproducibly avoid PA14 in choice assays. Individual replicates from *Kaletsky et al., 2020*, *Moore et al., 2021*, are shown for P0 worms obtained from P11 vs control training. (**B**) Individual replicates from (**A**) were pooled. (**C**) F2 animals from P0 worms trained on *E. coli* expressing the P11 small RNA reproducibly avoid PA14 in choice assays. Individual replicates from *Kaletsky et al., 2020*, *Moore et al., 2021*,

*Figure 3 continued on next page*

*Figure 3 continued*

are shown for F2 worms obtained from P11- vs control-trained grandmothers. (**D**) Individual replicates from (**A**) were pooled. (**A, C**) Each replicate is a separate experiment made up of 5–10 plates (average ~80 worms per plate). The mean difference (effect size) is shown as a Cumming estimation plot. Each mean difference is plotted as a bootstrap sampling distribution. Mean differences are depicted as dots; 95% confidence intervals are indicated by the ends of the vertical error bars. (**E**) Normalized RNA counts for P11 expression from PA14 under different bacterial growth conditions. Sequencing data and figure adapted from *Kaletsky et al., 2020*. Adjusted p-values from DESeq2. (**F**) P0 *meg-1;meg-3* mutants exhibit normal learned PA14 avoidance when trained on PA14 lawns, but (**G**) P0 *meg-1;meg-3* mutants do not exhibit learned PA14 avoidance when trained with PA14 sRNA. (**H**) P0 *nrde-1(gg66)* mutants exhibit normal learned PA14 avoidance when trained on PA14 lawns, (**I**) but are defective for PA14 avoidance learning when exposed to PA14 sRNA. (**J**) F2 *nrde-3* mutants from P0-PA14 lawn-trained grandmothers do not avoid PA14 compared to wild-type controls. (**K**) Wild-type P0 worms exposed to *E. coli*-expressing P11 avoid PA14, but *rde-3* mutants do not. (**B, D–K**) Box plots: center line, median; box range, 25–75th percentiles; whiskers denote minimum–maximum values. (**B, D**) Unpaired, two-tailed Student's t-test. (**F–K**) Two-way ANOVA with Tukey's multiple comparison's test. *p≤0.05, ***p≤0.001, ****p<0.0001, NS, not significant. Estimation graphics generated as described in *Ho et al., 2019*.

the lower temperature, and a 13.7-fold decrease in P11 when PA14 is grown in liquid rather than on plates, even at high temperatures. Therefore, it is clear that the growth of PA14 in non-25°C/plate-grown conditions may fail to elicit a P11-mediated avoidance response, because no P11 is expressed in PA14 under those conditions. Because *Gainey et al., 2024*, did not verify P11 expression in their bacterial cultures, it is not surprising that they also failed to observe P11-induced avoidance in the F2 generation in those experiments.

## TEI of learned avoidance requires components of RNA processing pathways

Previously, we showed that many components of small RNA uptake, RNA interference, small RNA processing pathways, and the germline are necessary for transgenerational inheritance of learned avoidance (*Moore et al., 2019*; *Kaletsky et al., 2020*). We find that the *meg-1;meg-3* double mutants, which perturb P granules and the germline (*Wang et al., 2014*), can avoid PA14 after lawn training (*Figure 3F*) but are defective for small RNA-induced learning (worms trained on *E. coli* with small RNA purified from PA14 added to the lawn) (*Figure 3G*). Similarly, NRDE-3, an Argonaute protein that functions in nuclear RNAi (*Guang et al., 2008*), is also required for small RNA-induced learned avoidance (*Figure 3H-J*). Like other mutants required for the small RNA TEI pathway, the P0 generation of *meg-3;meg-4* and *nrde-3* mutants still avoid PA14 after training on intact PA14 lawns (*Figure 3F and H*) through other avoidance pathways (innate immunity, metabolites, etc.), once again highlighting the independence of these different avoidance-inducing pathways. Only the small RNA-induced avoidance is transgenerational, i.e., F2s of wild-type worms continue to avoid PA14 after lawn training, but F2s of *nrde-3* mutants are defective for this inherited small RNA-induced avoidance (*Figure 3J*).

We also find that *rde-3/mut-2*, which encodes a ribonucleotidyltransferase that is involved in siRNA accumulation, piRNA-mediated silencing, and p(UG)ylation of small RNAs (*Chen et al., 2005*; *Shukla et al., 2020*; *Priyadarshini et al., 2022*) in epigenetic inheritance is also required for *E. coli* P11-induced learned avoidance (*Figure 3K*). Thus, components of small RNA transport, small RNA processing, RNAi function, germline components, chromatin factors, neuronal function, and the Cer1 retrotransposon (*Moore et al., 2021*) are required for the transgenerational inheritance of learned pathogen avoidance (*Figure 4A*).

## Discussion

*C. elegans'* physiological responses to PA14 and other pathogenic bacteria may allow the worms to increase their chances of survival. Intergenerational responses such as increased survival (*Burton et al., 2020*; *Pender et al., 2025*), changes to development (*Palominos et al., 2017*), and internal physical responses (*Singh and Aballay, 2019*; *Das et al., 2024*) likely also increase the possibility of the survival of their offspring. Switching behavioral responses from attraction to avoidance for a few generations may also increase the survival of the species (*Moore et al., 2019*) by reducing the chances of interactions with the pathogens. Of the many cues that induce avoidance, only the bacterial small RNA pathway induces transgenerational inheritance of learned pathogen avoidance, and this pathway is molecularly distinct from other pathways that induce avoidance only in the P0 or F1 generations (*Kaletsky et al., 2020*). That is, if PA14 is grown under conditions in which P11 is *not*

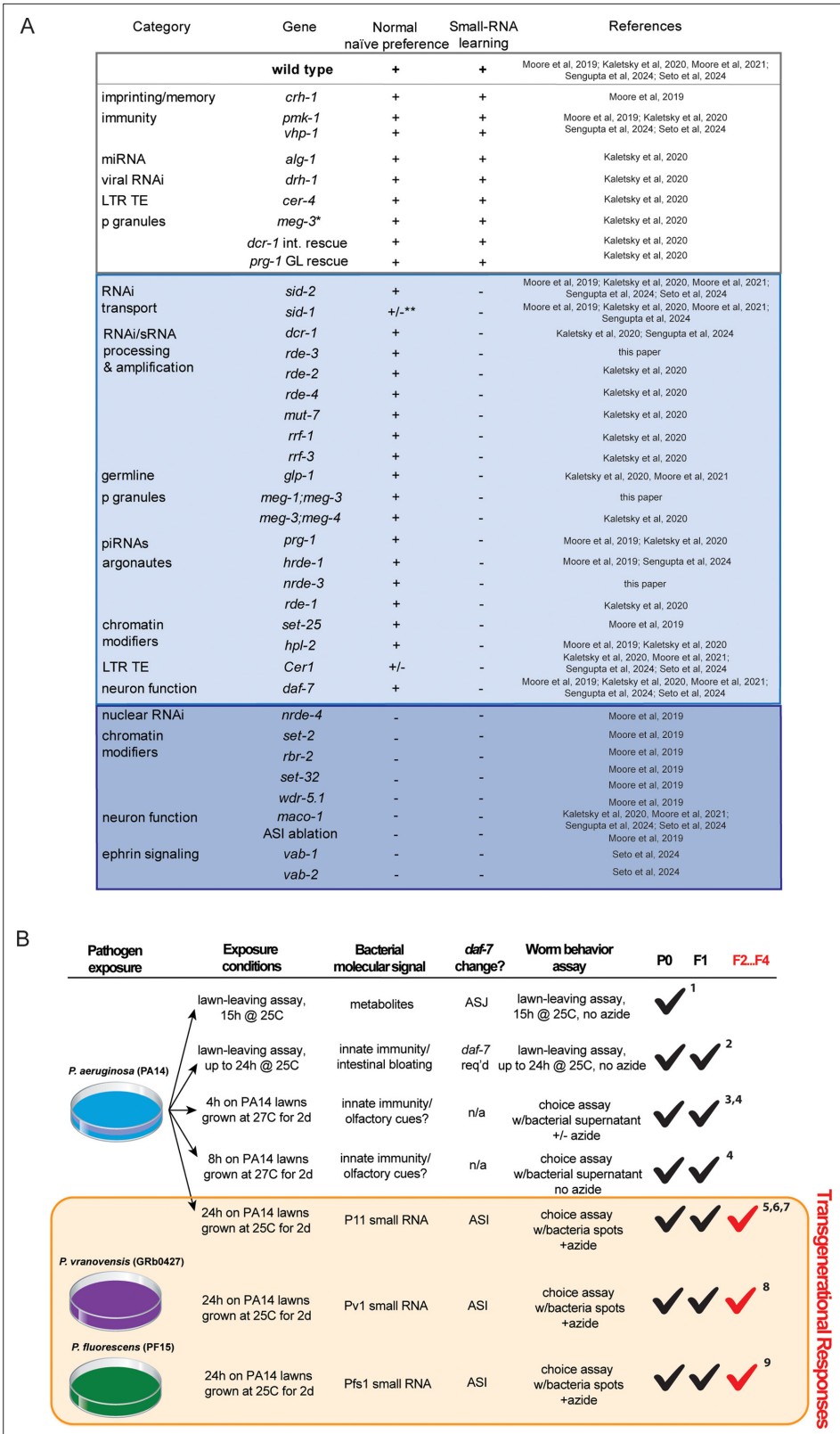

**Figure 4.** Summary of genes tested for small RNA-mediated pathogen avoidance and characterization of PA14 avoidance-based assays. (**A**) Summary of genes tested for naïve *Pseudomonas* preference behavior and small RNA-based learning and inheritance. (**B**) Table describing different assays measuring *Pseudomonas* learning and avoidance. *Pseudomonas* avoidance learning in the parental generation (**P0**), intergenerationally (**F1**), and

*Figure 4 continued on next page*

*Figure 4 continued*

transgenerationally (F2 through F4) is noted. References: 1, *Meisel et al., 2014*; 2, *Singh and Aballay, 2019*; 3, *Zhang et al., 2005*; 4, *Pereira et al., 2020*; 5, *Moore et al., 2019*; 6, *Kaletsky et al., 2020*; 7, *Moore et al., 2021*; 8, *Sengupta et al., 2024*; 9, *Seto et al., 2025*.

expressed, the bacteria may still induce P0 and even F1 avoidance due to the presence of olfactory, metabolite, and other cues, but avoidance in the F2 generation and beyond will be absent. Therefore, if F2 avoidance is not observed, it might be due to the lack of P11 expression in those conditions, and conversely, avoidance in the P0 and F1 conditions should not be misconstrued as positive evidence of P11 expression.

P11 sRNA is a regulator of PA14's nitrogen assimilation operon, which includes nitrite and nitrate reductases (NirB, NirD, and NasC) that control the production of ammonium (*Marogi et al., 2024*). Nitrogen assimilation appears to be critical for growth on surfaces at specific temperatures, including inside its host, *C. elegans* (*Marogi et al., 2024*). Nitrogen assimilation is an energetically costly process, and PA14 tightly regulates this operon via control of the P11 expression. Interestingly, loss of P11 reduces ammonia production and thus reduces *C. elegans'* chemoattraction to PA14 (*Marogi et al., 2024*); therefore, differences in *C. elegans* behavior when treated with PA14 vs ΔP11-PA14 mutants can be due to both the loss of the P11 small RNA itself and also due to the loss of ammonia production. Therefore, any growth conditions that result in disruption of P11 expression might also contribute to decreases in naïve attraction that Gainey et al. observed, i.e., if PA14 is grown under conditions where P11 sRNA is not expressed, not only would training on this bacteria not induce transgenerational inheritance of avoidance due to the lack of P11 signal, but these conditions might also decrease *C. elegans'* naïve attraction to this PA14, further reducing behavioral differences between naïve and trained chemotaxis indices. Thus, growth conditions that decrease P11 expression might have a large effect on both naïve and trained chemotaxis choice indices, and thus on the interpretation of these effects. Because P11 sRNA is both necessary and sufficient for the transgenerational inheritance of pathogen avoidance, and its expression level is acutely sensitive to both temperature and physical environmental conditions, it would be important to confirm P11 expression levels (e.g. via qPCR) in PA14 rather than assuming that P11 is constitutively expressed, particularly when attempting to report the lack of an F2 effect. Unfortunately, *Gainey et al., 2024* never carried out any tests of P11 sRNA in PA14, so it is impossible to now know how the PA14 growth conditions in any of their experiments might have affected P11 levels and subsequent *C. elegans* responses. Additionally, learned avoidance in the P0 and F1 generations has no relationship to the P11 pathway and therefore cannot be used as confirmation of P11 expression.

In addition to P11, we also previously found that *C. elegans* ingestion and processing of specific bacterial small RNAs produced by *P. vranovensis* (*Sengupta et al., 2024*) and PF15 (*Seto et al., 2025*) induce pathogen avoidance that lasts through the F4 generation. These specific small RNAs are Pv1 and Pfs1 (from *P. vranovensis* and PF15, respectively), and they each contain a stretch of either 17 (P11) or 16 (Pv1, Pfs1) nucleotides in predicted stem loops that are a perfect match to specific exons of the gene *maco-1* (P11, Pv1) or *vab-1* (Pfs1), which are then downregulated upon treatment with these pathogenic bacteria or these small RNAs. In fact, editing the small RNAs to disrupt the perfect match to *maco-1* or *vab-1* abrogates the learned avoidance (*Sengupta et al., 2024*; *Seto et al., 2025*), underscoring the requirement for the perfect match sequence between the bacterial small RNAs and their target *C. elegans* genes. MACO-1 is an ER resident protein that has been previously implicated in regulation of chemotaxis, the dauer developmental decision, and thermotaxis (*Arellano-Carbajal et al., 2011*; *Miyara et al., 2011*; *Neal et al., 2016*). *maco-1* downregulation upon PA14 treatment (25°C on plates) or upon treatment with *E. coli* expressing P11 (*E. coli*-P11) results in upregulation of *daf-7::gfp* in the ASI neurons (*Kaletsky et al., 2020*). Using qPCR, we confirmed the downregulation of *maco-1* in P0 through the F4 generations and observed its return to normal levels in the F5 generation (*Sengupta et al., 2024*), corresponding with avoidance behavior. Furthermore, mutants of *maco-1* exhibit high naïve avoidance of PA14 and high levels of *daf-7::gfp* expression in the ASI prior to training, and because of this high naïve avoidance, they are unable to learn further avoidance upon training on PA14 (*Kaletsky et al., 2020*). P0 worms treated with *maco-1* RNAi exhibit transgenerational inheritance of learned avoidance, as well (*Kaletsky et al., 2020*). The downregulation of *vab-1* upon PF15 or Pfs1 sRNA treatment also results in the downregulation of *maco-1* expression levels (as

shown by RNA-seq in P0 mothers), suggesting that *vab-1* is upstream of *maco-1*; qPCR confirmed that *maco-1* is downregulated in the F2 generation of *E. coli*-Pfs1-trained mothers as well (**Seto et al., 2025**). Therefore, the transgenerational bacterial small RNA–*C. elegans* avoidance mechanism induced by at least three different *Pseudomonas* species through three different small RNAs requires the downregulation of *maco-1*, which results in *daf-7* changes in the ASI; perfect matches between bacterial small RNAs and their *C. elegans* target genes are required for the F2–F4 heritable shift from attraction to avoidance. Thus, the role of *maco-1* in this pathway has been confirmed through multiple avenues of experimentation.

Through extensive mutant analyses (testing of almost 40 mutants over the course of six studies), we have found that this sRNA mechanism of transgenerational inheritance of learned pathogen avoidance requires a wide range of machinery involved in bacterial small RNA uptake, RNA processing, signaling, and neuronal behavioral execution (*Figure 4A*). These genes include RNA interference components, Argonautes, and RNA-dependent polymerases; germline components such as P-granule, piRNA components, COMPASS, and chromatin modifiers; the Cer1 Ty3 Gypsy retrotransposon; and neuronal genes (*maco-1*/macoilin, *vab-1*/Ephrin receptor, *vab-2*/Ephrin) as well as the *daf-7*/TGF-beta ligand in the ASI neuron (*Figure 4A*). We also ruled out several genes that function in other forms of pathogen avoidance, such as imprinting, innate immunity signaling, microRNA, and viral RNAi processors, and other transposable elements, further distinguishing the bacterial small RNA TEI pathway from other avoidance mechanisms that may only function in the P0 or F1 generations (*Figure 4A and B*). Through tissue-specific rescue experiments, we also showed that *dcr-1* functions in the intestine and *prg-1* is necessary in the germline for small RNA-mediated learned pathogen avoidance. Our studies of other bacterial species have identified not only additional bacterial small RNAs that are 'read' by *C. elegans* (**Sengupta et al., 2024**; **Seto et al., 2025**), but also have identified additional *C. elegans* gene targets that function in the pathway (e.g. *vab-1*, which regulates *maco-1*; **Seto et al., 2025**). It should be noted that in each of these mutant experiments, wild-type worms were tested as controls; therefore, our wild-type naïve attraction and learned avoidance results have been replicated many times over the course of these six studies.

## Conclusions

Here, we have investigated two experimental factors that might affect investigators' interpretation of PA14 training-induced avoidance choice assays: the expression of P11 sRNA in PA14 and proper execution of the choice assay itself. It would be incorrect to assume that because F2 avoidance is not demonstrated under all PA14 growth conditions that '*the published model is unlikely to be relevant in a natural environment*' (**Gainey et al., 2024**). Instead, our results suggest the opposite: that the worms' ability to interpret P11 expression is due to P11's importance in regulating nitrogen fixation under biofilm-inducing conditions that also induce a pathogenic state that is important for *C. elegans* to remember over several generations. *C. elegans* appears to have evolved a mechanism to use P11 and other specific small RNAs as biomarkers of future infection, as the presence of those specific small RNAs correlates well with pathogenesis – and the specific small RNA detected by *C. elegans* in each case might be tightly correlated with the pathogenic potential of that particular *Pseudomonas* species. Furthermore, the fact that we found that other *Pseudomonas* species, including *Pseudomonas* from the worm's microbiota, use the same bacterial sRNA/*C. elegans* target gene mechanism indicates that this natural example of trans-kingdom signaling and transgenerational avoidance is highly conserved. In those bacteria, the key small RNAs appear to be constitutively expressed, and thus even more likely to be encountered by *C. elegans* upon ingestion. Together, these data suggest that the transgenerational inheritance of small RNA-induced learned bacterial avoidance is likely to be widespread in the wild, indicating its likely importance as a mechanism to promote long-term survival of *C. elegans* in its natural habitat.

## Materials and methods
### General worm maintenance
Worm strains were maintained at 20°C on high growth medium (HG) plates (3 g/L NaCl, 20 g/L bacto-peptone, 30 g/L bacto-agar in distilled water, with 4 mL/L cholesterol [5 mg/mL in ethanol], 1 mL/L 1 M CaCl$_2$, 1 mL/L 1 M MgSO$_4$, and 25 mL/L 1 M potassium phosphate buffer [pH 6.0] added to

molten agar after autoclaving) on OP50 using standard methods. The following worm strains were provided by the *C. elegans* Genetics Center (CGC): JH3229, *meg-1(vr10); meg-3(tm4259)* X; YY158, *nrde-3(gg66)* X; WM30, *mut-2/rde-3(ne298)* I.

### *E. coli* and PA14 culture conditions

OP50 was provided by the CGC. PA14 was a gift from Z. Gitai. OP50 and PA14 were grown in liquid cultures in LB at 37°C with shaking. *E. coli* strains expressing PA14 sRNA were cultured overnight in Luria broth supplemented with 0.02% arabinose wt/vol and 100 µg/mL carbenicillin. Overnight cultures of bacteria were diluted in LB to an optical density at 600 nm (OD 600)=1 and used to fully cover nematode growth medium (NGM) (3 g/L NaCl, 2.5 g/L bacto-peptone, 17 g/L bactoagar in distilled water, with 1 ml/L cholesterol [5 mg/mL in ethanol], 1 mL/L 1 M $CaCl_2$, 1 mL/L 1 M $MgSO_4$ and 25 mL/L 1 M potassium phosphate buffer [pH 6.0] added to molten agar after autoclaving) plates. For the preparation of control *E. coli* and *E. coli* expressing the PA14 P11 small RNA, bacteria were seeded on NGM plates supplemented with 0.02% arabinose and 100 µg/mL carbenicillin. All plates were incubated for 2 days at 25°C, unless specified otherwise (in separate incubators for control and pathogen-seeded plates). OP50 and PA14 small RNAs were collected as described in *Kaletsky et al., 2020*.

### Aversive learning assay

Worm preparation for training: synchronized L4 worms were washed off plates using M9 and left to pellet on the bench top for approximately 5 min. Then, 5 µL of worms were placed onto sRNA-spotted training plates, and 10 µL or 40 µL of worms were plated onto OP50 or *E. coli* expressing control/ P11, or PA14-seeded training plates, respectively. Worms were incubated on training plates at 20°C in separate containers for 24 hr. After 24 hr, worms were washed off plates using M9 and washed an additional three times to remove excess bacteria.

Choice plates and naïve day 1 worms were prepared as in *Moore et al., 2019*, and *Seto et al., 2025*. On the day of the assay, choice assay plates were left at room temperature for 1 hr before use. For standard (sodium azide added) experiments, 1 µL of 1 M sodium azide was spotted onto each respective bacteria spot to be used as a paralyzing agent during choice assay and preserve first bacterial choice. For conditions without azide, no azide was added.

Worms were washed off training plates in 2 mL M9 into 1.5 mL tubes and allowed to pellet by gravity. Worms were washed two to three additional times in M9. Using a wide orifice pipet tip, 5 µL of worms (approximately 50–200 worms) were spotted at the bottom of the assay plate, midway between the bacterial spots. Plates were incubated at room temperature for 1 hr before moving plates to 4°C. Plates were left at 4°C for ~1 hr before manually counting the number of worms on each bacterial spot.

### Meta-analysis

We collected all examples of N2 naïve chemotaxis (PA14 vs OP50 choice assays) prior to training, N2 P0 and F2 choice assays after P0 training on PA14 vs OP50, and P0 and F2 choice assays after P0 training on *E. coli*-P11 vs control *E. coli* from *Moore et al., 2019*, *Kaletsky et al., 2020*, *Moore et al., 2021*. These data are collected in *Supplementary file 2*. Multiple two-group estimation plots were generated for side-by-side comparison of multiple mean differences across independent experiments. Each mean difference is plotted as a bootstrap sampling distribution.

## Additional information

### Funding

| Funder | Grant reference number | Author |
|---|---|---|
| National Institute of General Medical Sciences | DP1GM119167 | Coleen T Murphy |
| Ford Foundation Fellowships | | Renee Seto |

| Funder | Grant reference number | Author |
| --- | --- | --- |
| National Institutes of Health | T32GM007388 | Rebecca S Moore<br>Renee Seto |

The funders had no role in study design, data collection and interpretation, or the decision to submit the work for publication.

## Author contributions

Rachel Kaletsky, Conceptualization, Formal analysis, Supervision, Validation, Investigation, Visualization, Methodology, Writing – original draft, Writing – review and editing; Rebecca S Moore, Titas Sengupta, Investigation, Methodology, Writing – review and editing; Renee Seto, Validation, Investigation, Visualization, Methodology, Writing – original draft, Writing – review and editing; Borja Ceballos-Llera, Investigation; Coleen T Murphy, Conceptualization, Supervision, Funding acquisition, Writing – original draft, Project administration, Writing – review and editing

## Author ORCIDs

Rachel Kaletsky (ID) https://orcid.org/0000-0002-7176-9154
Rebecca S Moore (ID) https://orcid.org/0000-0003-3155-3436
Titas Sengupta (ID) https://orcid.org/0000-0002-7228-719X
Coleen T Murphy (ID) https://orcid.org/0000-0002-8257-984X

Reviewer #1 (Public review): https://doi.org/10.7554/eLife.105673.2.sa1
Reviewer #2 (Public review): https://doi.org/10.7554/eLife.105673.2.sa2
Reviewer #3 (Public review): https://doi.org/10.7554/eLife.105673.2.sa3

---

# Additional files

## Supplementary files

Supplementary file 1. Summary data compiled from *Moore et al., 2019*, *Kaletsky et al., 2020*, and *Moore et al., 2021*.

Supplementary file 2. *Figure 3* worm counts from choice assays.

MDAR checklist

## Data availability

All data generated or analyzed during this study are included in the manuscript and supporting files; source data files have been provided for all figures in *Supplementary files 1 and 2*.

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
