## [Editor Report · eLife Assessment]

This **fundamental** study concerns a model for transgenerational epigenetic inheritance, the learned avoidance by *C. elegans* of the PA14 pathogenic strain of *Pseudomonas aeruginosa*. The authors test the impact of procedural alterations made in another study, by Gainey et al., which claimed that transgenerational inheritance in this paradigm lacks robustness, despite this observation having been reported in multiple papers from the Murphy lab. The authors of the present study show that by following a non-standard avoidance protocol, Gainey et al. likely biased their measurements in a way that made it hard to observe learned avoidance. The authors also highlight the importance of bacterial growth conditions, showing that expression of the trigger molecule, the bacterial P11 RNA, which is necessary and sufficient to drive the transgenerational inheritance of the avoidance phenotype, is influenced by temperature. As expression of P11 was not verified by Gainey et al., this provides another explanation for the inability to observe transgenerational epigenetic inheritance. Together, the authors provide **compelling** and powerful arguments that the original phenomenon is robust and that it can be reproduced in the Murphy lab by following their original protocol precisely, including the use of azide to immobilize the worms at the food source. Overall, this study not only provides guidance for investigators in this experimental paradigm, but it also provides additional understanding of the differences between naïve preference, learned preference, and transgenerational epigenetic inheritance. The present study is therefore of broad interest to anyone studying genetics, epigenetics, or learned behavior.

---

## [Referee Report · Reviewer #1 (Public review)]

Summary:

The manuscript from Kaletsky et al is a response to a paper recently published by Craig Hunter's group (Gainey et al 2024). The Murphy lab has previously shown that learned avoidance of *C. elegans* to PA14 can be transmitted through four generations. In a series of detailed studies, they defined the mechanism of this transgenerational epigenetic inheritance (TEI), identifying both PA14 and *C. elegans* factors required for this effect (Moore et al., 2019, Kaletsky et al., 2020; Moore et al., 2021). PA14 produces a small RNA, P11, that is necessary and sufficient for transgenerational epigenetic inheritance of avoidance behaviour in *C. elegans*. In the worm, P11 decreases maco-1 expression, which in turn regulates daf-7.

In the study by Gainey et al (eLife 2024), the authors report their attempt at replicating the original findings of the Murphy lab using a modified experimental setup. The Gainey study observed avoidance of PA14 and upregulation of daf-7::GFP in the F1 progeny of trained parents, but not in subsequent generations. Importantly, although they examined a number of different deviations of the protocol, they did not repeat the original experiment using the exact protocol outlined in the Moore or Kaletsky papers. Nevertheless, the authors concluded that "this example of TEI is insufficiently robust for experimental investigations".

The manuscript by Kaletsky et al. attempts to provide an explanation as to why Gainey et al., were unable to observe transgenerational avoidance of PA14. They identify two discrepancies in the methodology used between the two studies and examine the possible impacts of these.

One of the primary differences in protocols between the two papers is how avoidance is measured. The Murphy group uses the traditional method of adding azide to bacterial spots on the choice plates to trap worms once they have come close to the food spot. The animals are on the plate for 1 hour but most have likely been immobilized before this time point. Gainey et al. omit the azide and instead shift animals to 4C after 30-60 minutes of exposure to immobilize the worms for counting. Kaletsky et al show that the choice of assay has a significant impact on measuring attraction and avoidance.

While Gainey et al., assert that the addition of azide had no discernable effect on the choice assay results, these data are not shown in their paper. Kaletsky et al. test these conditions head-to-head with the same 1 hour exposure time, showing that with azide, the initial response to PA14 in untrained worms is attraction. By contrast, in the absence of azide, when cold temperature is used to immobilize the worms , the response recorded is aversion to PA14. The choice assay generated by Kaletsky et al without azide is consistent with the choice assays in untrained worms shown in the Gainey paper, demonstrating that this is likely one factor that contributed to the different outcomes reported in the Gainey paper.

Kaletsky et al. propose that learned aversion to PA14 may be occurring within the 1-hour exposure time when worms are not trapped in their initial decision with the use of azide. This is consistent with previous findings from another group (Ooi and Prahlad 2017), showing that 45 minutes of exposure is sufficient to overcome the attraction to PA14 and shift to avoidance of PA14. Importantly, the Gainey paper notes exposure times between 30 and 60 minutes before shifting worms to 4C to count, this window may have generated additional variability between assays.

The second possibility explored by Kaletsky et al. is that the expression of P11 differed between the studies. Because P11 is required for TEI, differences in P11 expression is a reasonable explanation for different observations between studies. Unfortunately, in the Gainey study, P11 levels were not measured; it is therefore not possible to know whether low or absent levels of P11 explain the inability to observe TEI. Nevertheless, Kaletsky et al. test the potential for changes in one growth condition, temperature, to influence the production P11. Indeed, the expression of P11 differs in PA14 grown at different growth temperatures, providing an additional explanation for the discrepancies.

While it is possible that temperature is the culprit, it may be another culture condition or media component suppressing P11 expression. Nevertheless, the fact that expression of P11 can so easily be modified demonstrates that P11 expression is not immune to differences in culture conditions. Given its role in nitrogen fixation, I would be surprised if it was not regulated by environmental conditions. Differences in iron content between media batches are notorious for altering bacteria phenotypes. Although outside the scope of this study, with the connection to biofilm formation, I would be curious if iron levels had an impact on P11 expression. All in all, the data highlight the fact that P11 levels should be measured if TEI is not seen.

Strengths:

Overall, this is an excellent study that has provided additional understanding of the difference between naïve preference and TEI and provides guidance for investigators in replicating TEI experiments. The manuscript is very well written and provides additional understanding regarding the replication of TEI in response to *P. aeruginosa*.

The manuscript provides an important discussion about differences in methodology and how they might reflect specific biology. Many examples of experimental deviations that have large impacts have simple biological explanations. I believe the authors have done an excellent job making this point.

Weaknesses:

None noted.

---

## [Referee Report · Reviewer #2 (Public review)]

In addition to the study by Kaletsky et al. (2025), I read the bioRxiv and eLife versions, as well as the eLife reviewer comments, for Gainey et al. (2024), to which Kaletsky et al. respond.

Kaletsky et al. provide detailed, rigorous, and reproducible protocols and results. The authors point out the critical methods that the Hunter group failed to follow/confirm (e.g. azide to paralyze animals during pathogenic learning/memory assays; the expression of the P11 small RNA that is both necessary and sufficient for TEI of avoidance behavior; a single condition for training - PA14 grown on plates at 25°C and training at 20°C for 24 hr - that the Hunter lab did not follow and could not reproduce). The Kaletsky et al. response is evidence-based, fair, level-headed and unbiased, which is in contrast to the Gainey et al. paper.

Reading the eLife review of Gainey et al., I note that the reviewers repeatedly pointed out that authors did not follow published protocols by the Murphy lab.

Public response by Gainey et al. to Reviewer 2: "It remains possible that we misunderstood the published Murphy lab protocols, but we were highly motivated to replicate the results so we could use these assays to investigate the reported RNAi-pathway dependent steps, thus we read every published version with extreme care."

Public response by Gainey et al. to Reviewer 3: "We agree that our study was not exhaustive in our exploration of variables that might be interfering with our ability to detect F2 avoidance."

Gainey et al. provide reasons/excuses for why they did not follow published methods - notably their subjective decision to exclude the paralyzing agent sodium azide from their choice assays, but their abstract reads "We conclude that this example of transgenerational inheritance lacks robustness." I strongly disagree with this conclusion.

---

## [Referee Report · Reviewer #3 (Public review)]

A recent bioRxiv paper from Craig Hunter's lab (Gainey et al. 2024) puts into question several manuscripts that report that pathogen avoidance by the nematode *C. elegans* to the pathogenic bacteria, *Pseudomonas aeruginosa*, for several generations after initial exposure is not robust nor repeatable. From the Hunter lab publication, the authors tried to eliminate genetic drift of the pathogenic bacterial strains and *C. elegans*, as well as several experimental conditions, including assay temperature conditions and the effect of light.

The papers (Moore et al. 2019, Kaletsky et al. 2020, Moore et al. 2021 and Sengupta et al. 2024) that the Gainey et al. manuscript brings into question discovered that *Pseudomonas aeruginosa* can produce a small RNA (sRNA), P11, that is necessary and sufficient for pathogen avoidance of the future generation of *C. elegans* (up to F4 generation). The Gainey et al. manuscript does not assess the status of P11 production in their work.

Here, the Murphy group has made several new discoveries that highlight the differences with the work performed in the Hunter lab. One, the assay used to test attraction and avoidance of *C. elegans* for pathogenic bacteria differs amongst the two groups. In the Murphy lab papers, and many others in this field, the assay is established whereby worms can decide between spots of non-pathogenic bacteria (*E. coli*) or pathogenic (*P. aeruginosa*) on a single plate separated by a few centimeters. Also included in each spot is an aliquot of NaN3 to freeze the animals upon entry into their first bacterial choice. *C. elegans* will initially choose the pathogenic bacteria as its first choice and then learn to avoid the pathogenic spot thereafter. Therefore, establishing this first baseline attraction point is essential for determining future avoidance events. The Hunter lab did not use NaN3 and instead relied upon moving plates to 4°C to slow the worm's movements to count the population. Furthermore, the Hunter lab allowed the "choice" to proceed for an hour before moving to 4°C, making capture of the initial attraction phase of the choice assay difficult to discern since the worms could move freely from their initial choice due to the lack of the paralyzing NaN3.

The second major advance that the Murphy group has found is that the growth of *P. aeruginosa* prior to being used for the choice assay is critical. Growth on plates at 25°C, but not 20°C on plates or in liquid at 37°C, can produce the transgenerational inheritance of pathogen avoidance. Interestingly, P11 is only produced by *P. aeruginosa* at 25°C grown on plates. The Hunter group grew the Pseudomonas bacteria at 37°C in liquid with gentle shaking and then spotted onto assay plates followed by growth for 2 days at 25°C and then equilibrated to room temperature before the choice assay. The Hunter lab did not check the status of P11 production in any of their experiments.

The results from the Murphy group are solid and they go on to find genetic requirements in *C. elegans* required for the transgenerational response to *P. aeruginosa* and P11. Furthermore, they repeat their results with additional members of the Pseudomonas clade and find the same transgenerational avoidance response and new sRNAs responsible for the avoidance response to the newly tested Pseudomonas members.

Overall, the discrepancies between the Hunter work and the numerous papers for the Murphy group would tend to complicate this area of research. However, this eLife paper plainly illustrates the straightforward nature of the experimental setup and reconfirms the necessary and sufficient nature of P11 in orchestrating the multigenerational response to pathogenic Pseudomonas. It appears that ensuring the production of P11 from the Pseudomonas culture and ensuring that the assay captures the initial bacterial choice are essential to observe the transgenerational inheritance of the avoidance phenotype.